# Estimation of the incubation period of COVID-19 in Vietnam

**Long V. Bui**[1]*, **Ha T. Nguyen**[2], **Hagai Levine**[3], **Ha N. Nguyen**[4], **Thu-Anh Nguyen**[5,6], **Thuy P. Nguyen**[7], **Truong T. Nguyen**[8], **Toan T. T. Do**[9], **Ngoc T. Pham**[10], **My Hanh Bui**[9,11]

**1** Center for Research - Consulting and Support of Community Heath, Hanoi, Vietnam, **2** 108 Military Central Hospital, Hanoi, Vietnam, **3** The Braun School of Public Health and Community Medicine, Hadassah - Hebrew University, Jerusalem, Israel, **4** People's Police Academy, Hanoi, Vietnam, **5** Woolcock Institute of Medical Research, Hanoi, Vietnam, **6** Faculty of Medicine and Health, University of Sydney, Camperdown, New South Wales, Australia, **7** Menzies Institute for Medical Research, University of Tasmania, Hobart, Australia, **8** Ministry of Science and Technology, Hanoi, Vietnam, **9** Hanoi Medical University, Hanoi, Vietnam, **10** 198 Hospital, Hanoi, Vietnam, **11** Hanoi Medical University Hospital, Hanoi, Vietnam

* buimyhanh@hmu.edu.vn

## Abstract

### Objective

To estimate the incubation period of Vietnamese confirmed COVID-19 cases.

### Methods

Only confirmed COVID-19 cases who are Vietnamese and locally infected with available data on date of symptom onset and clearly defined window of possible SARS-CoV-2 exposure were included. We used three parametric forms with Hamiltonian Monte Carlo method for Bayesian Inference to estimate incubation period for Vietnamese COVID-19 cases. Leave-one-out Information Criterion was used to assess the performance of three models.

### Results

A total of 19 cases identified from 23 Jan 2020 to 13 April 2020 was included in our analysis. Average incubation periods estimated using different distribution model ranged from 6.0 days to 6.4 days with the Weibull distribution demonstrated the best fit to the data. The estimated mean of incubation period using Weibull distribution model was 6.4 days (95% credible interval (CrI): 4.89–8.5), standard deviation (SD) was 3.05 (95%CrI 3.05–5.30), median was 5.6, ranges from 1.35 to 13.04 days (2.5th to 97.5th percentiles). Extreme estimation of incubation periods is within 14 days from possible infection.

### Conclusion

This analysis provides evidence for an average incubation period for COVID-19 of approximately 6.4 days. Our findings support existing guidelines for 14 days of quarantine of persons potentially exposed to SARS-CoV-2. Although for extreme cases, the quarantine period should be extended up to three weeks.

**Data Availability Statement:** Data of all confirmed COVID-19 cases in Vietnam were made publicly available on the official website of Ministry of Health of Vietnam (ncov.moh.gov.vn) Data for analysis and R codes are available on public

repository https://github.com/longbui/
Covid19IncubVN.

**Funding:** The authors received no specific funding
for this work.

**Competing interests:** The authors have declared
that no competing interests exist.

## Introduction

COVID-19 caused by SAR-CoV-2 has been declared a global pandemic by WHO on March 11
2020 [1]. As a newly emerged disease, little have we known about the incubation period for
COVID-19, for which patients have no clinical symptom but some infected persons can be
contagious [2]. Understanding the distribution of incubation period is important to estimate
the potential spreading of the SARS-CoV-2, and to determine an optimal duration of quaran-
tine. The incubation period is identified by the interval time between exposure to source of an
infectious disease and the onset of the first clinical symptoms. As the incubation period is
known, we would be able to model the current and future of the pandemic scale, and to evalu-
ate the effectiveness of intervention strategies, therefore, would be able to act swiftly several
intensive public health strategies for infectious diseases [3].

Since the first COVID-19 case found in Vietnam on 23 January 2020, as of 13 April 2020,
there are 265 confirmed cases all over the country. Of those, two-thirds of the patients have
recovered, and no death has been reported. The pandemic situation in Vietnam is significantly
different from that in other countries. Vietnam has implemented strong measures against
the SARS-CoV-2 transmission at an early stage. However, the country has limited resources,
hence understanding the time of intensive monitoring will be beneficial to plan for effective
measures to minimize the risk for hidden SARS-CoV-2 infection. While estimations of incuba-
tion period of SARS-CoV-2 have been conducted elsewhere in the world [4–13], there is no
estimation of this parameter using data from Vietnam, instead, efforts were only focusing on
describing pattern of COVID-19 pandemic in Vietnam [14–16]. This study aims to estimate
the incubation period of Vietnamese confirmed COVID-19 cases.

## Methods

### Data sources and collection

We used secondary data that publicly available from Viet Nam Ministry of Health (MoH).
According to the report flow of the Ministry of Health and National Steering Committee for
COVID-19 response, information of all laboratory-confirmed COVID-19 cases in Vietnam,
which include patient number, travelling history, contact tracing and clusters, date of labo-
ratory-confirmed, date of assertion, date and status of discharge was collected by the local
centers for disease control and hospitals where patients were being admitted. The informa-
tion then was reported daily to the Ministry of Health and officially made publicly available
on the website http://ncov.moh.gov.vn [16]. Additionally, the clusters of COVID-19 in a
northern province of Vietnam have been described by Thanh et al. [17]. Two researchers
independently reviewed the full text of each case report of confirmed COVID-19 cases in
Vietnam from 23 January 2020 to 13 April 2020, and entered data into a standardized case
reporting form to establish a database for this study. Any discrepancies in data extraction
were resolved by discussion between two researchers and facilitated by a third researcher to
reach consensus. Previous studies also used the same approach to generate data sources for
analysis [15, 16].

To assure the reliability of analysis, we selected only confirmed COVID-19 cases who are
Vietnamese and locally infected with available data on date of symptom onset (including fever,
cough, and shortness of breath) and clearly defined window of possible SARS-CoV-2 expo-
sure. This window period is defined as the date range between the earliest possible exposure,
which is the first contact with confirmed cases and the latest exposure, which is the most recent
contact with confirmed cases in the clusters.

## Statistical analysis

We assumed that the moment of infection occurred between the interval of possible SARS-CoV-2 exposure. The distribution of incubation period was estimated using maximum likelihood where the likelihood function of each case in the dataset was a single interval-censored by three parametric forms with Hamiltonian Monte Carlo method for Bayesian Inference: Weibull distribution, the Gamma distribution and the Lognormal distribution. Non-informative positive prior for the parameters of the three distributions were specified. Leave-one-out Information Criterion (LooIC) was used to assess the performance of three models. The differences of LooIC larger than two were considered as statistically significant [18]. Mean, median and posterior Credible Interval (CI) for each distribution were also estimated. Statistical analysis was conducted by rstan [19] in R 3.6.4 [20]. Data for analysis and R codes are available on public repository https://github.com/longbui/Covid19IncubVN.

## Ethical statement

Research only aims to protect and improve the quality of treatment, not for any purpose. All details about patients' information is confidential, therefore the Institutional Review Board and requirement for informed consent was waived.

## Results

From 23 Jan 2020 to 13 Apr 2020, a total of 265 positive-confirmed SARS-CoV-2 was reported in Vietnam. Of those, 38.5% were locally infected, 58.2% were female. The mean age was 36 years old, median age was 31 years old. There were 19 confirmed COVID-19 cases found between 23 January to 13 April 2020, who met the inclusion criteria. These include being Vietnamese nationality, locally infected with SARS-CoV-2, and had completed epidemiological data on exposure interval and date of symptom onset. Of those, 7 were male and 12 were female. The mean age was 38.47, median age was 37 (interquartile range: 25–50). There was no statistical difference in mean age and sex distribution between sub-group for analysis and the entire group ($p < 0.05$).

Average incubation periods estimated using different distribution model ranged from 6.0 days to 6.4 days (Table 1). According to the LooIc of proposed models, the Weibull distribution demonstrated the best fit to the data. The estimated mean of the incubation period using Weibull distribution model was 6.4 days (95% CI 4.89–8.5), the standard deviation (SD) was 3.05 (95%CI 3.05–5.30). The Gamma distribution model fitted significantly poorer than the Weibull model. The estimated mean of the incubation period of this model was 6.07 days (95% CI: 4.64–8.00), with the SD of 2.90 days (95% CI 1.91–4.80), The lognormal distribution showed the poorest fit to the data, the incubation period was estimated to be 6.4 (95% CI 4.6–10.2).

Estimation of median of incubation period using different models are presented in Table 2. Fig 1 shows the estimation distribution of incubation periods using Weibull model, with estimated median of 6.1 days, ranging from 1.4 to 13.0 days (2.5th to 97.5th percentiles). Detail estimated possible moment of infection with Weibull distribution is illustrated in Fig 2. The

**Table 1. Mean and SD of the estimated incubation period for confirmed Vietnamese COVID-19 cases between 23 Jan 2020 to 13 Apr 2020.**

| Distribution | Mean (days) | | SD (days) | | LooIC |
|---|---|---|---|---|---|
| | Estimate | 95% CI | Estimate | 95% CI | |
| Weibull | 6.4 | 4.9–8.5 | 3.0 | 2.0–5.4 | 103.9 |
| Gamma | 6.1 | 4.7–8.0 | 2.9 | 1.9–4.8 | 106.2 |
| Lognormal | 6.4 | 4.6–10.2 | 5.8 | 3.5–11.2 | 106.7 |

**Table 2. Percentiles of the estimated incubation period for confirmed Vietnamese COVID-19 cases between 23 Jan 2020 to 13 Apr 2020 with Weibull, Gamma and Lognormal distribution.**

| Percentiles | Incubation period distribution (days) | | | | | |
|---|---|---|---|---|---|---|
| | Weibull | | Gamma | | Lognormal | |
| | Estimate | 95% CI | Estimate | 95% CI | Estimate | 95% CI |
| 2.5th | 1.4 | 0.4–2.7 | 1.8 | 0.7–33.0 | 1.5 | 0.6–2.5 |
| 5th | 1.9 | 0.7–3.4 | 1.0 | 2.2–3.4 | 1.8 | 2.0–3.5 |
| 50th | 6.1 | 4.4–8.0 | 5.6 | 4.2–7.3 | 5.2 | 3.6–7.3 |
| 95th | 11.9 | 9.10–12.0 | 11.5 | 8.6–16.7 | 14.9 | 9.7–30.2 |
| 97.5th | 13.0 | 9.1–20.7 | 13.0 | 9.6–19.4 | 18.2 | 11.3–40.8 |
| 99th | 14.4 | 10.6–24.0 | 14.7 | 10.7–22.8 | 23.0 | 14.4–58.0 |

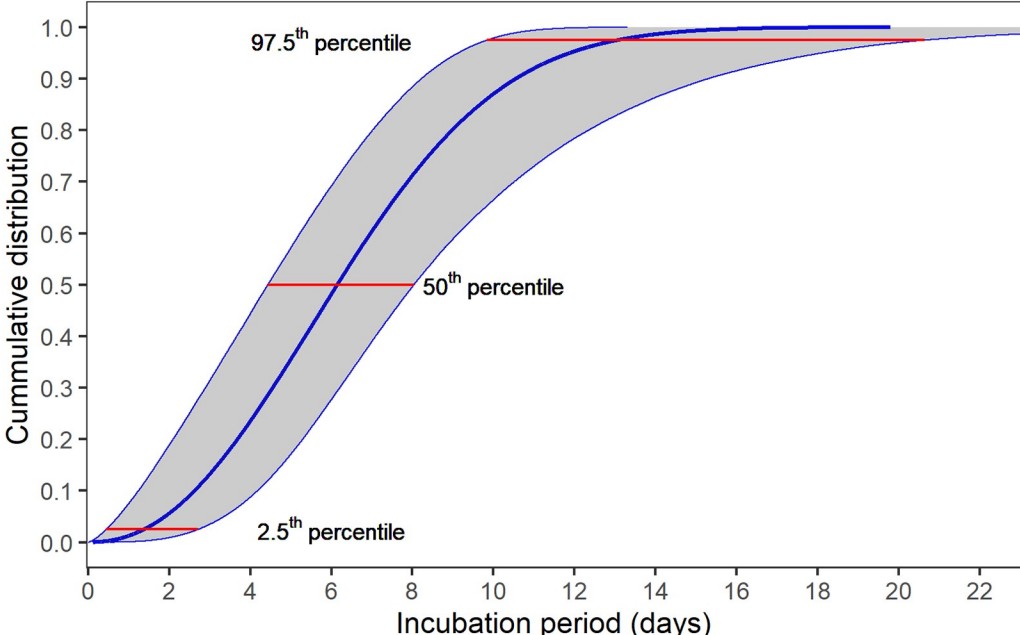

**Fig 1. The cumulative density function of the estimated Weibull distribution for incubation period of Vietnamese confirmed COVID-19 cases from 23 Jan 2020 to 13 Apr 2020.**

results suggest a large variation in incubation periods among patients but it was most likely to fall in the period within 14 days of infection.

## Discussion

We estimated the incubation period of Vietnamese confirmed COVID-19 cases from public reported data with three parametric models, including Weibull, Gamma and Lognornal distribution. The Weibull distribution proved to be best fit to the data. Our estimation results are similar or higher than most results from published literature (Table 3). Our estimation of 6.4 days for mean incubation period is similar to the estimation of Backer et al. [4]. The estimated mean in our study is higher in comparison with that of other studies, which ranged from 3 to 5.6 days [5–8, 10]. Our estimation is shorter than several studies. Leung et al. [9] estimated incubation periods of 7.2 days among local residents of Wuhan. Studies of Kong [11], Tindale et al. [12] and Qin et al. [13] estimated incubation periods around 8–9 days. When comparing

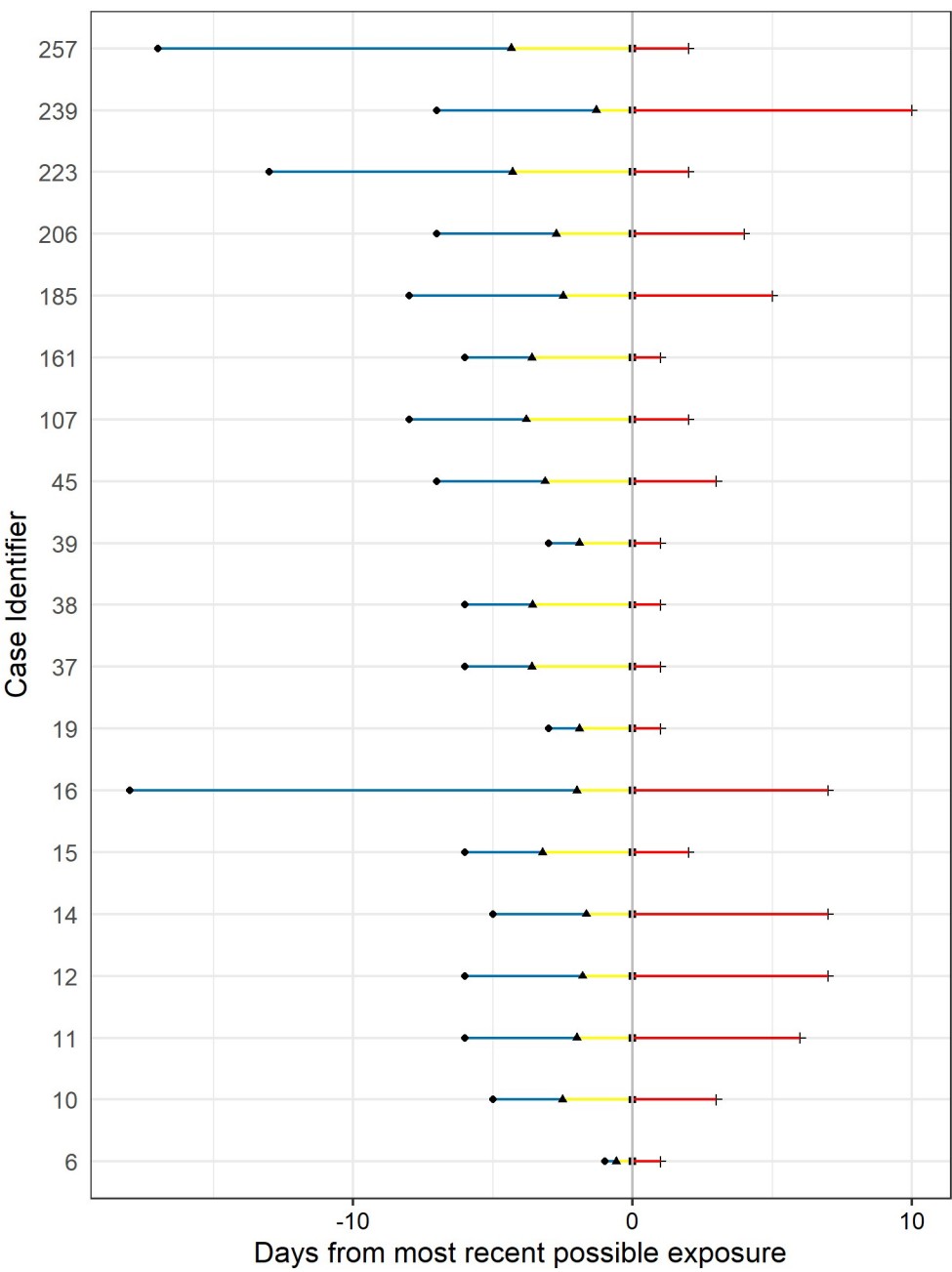

**Fig 2. Estimated possible moment of infection of Vietnamese confirmed COVID-19 cases from 23 Jan 2020 to 13 Apr 2020 with Weibull distribution.** (Cases were numbered by the order of official assertion of confirmed cases on the website of the Ministry of Health of Vietnam).

the estimated median and its range with other study, we saw a similar trend where our estimations are similar or higher than most studies where median estimation was available [4, 5, 7, 8]. Nonetheless, extreme estimation of incubation periods (95th or 97.5th percentiles) in studies are well within 14 days from possible infection [4, 5, 7–13].

**Table 3. Comparison of estimated incubation periods for SARS-COV-2.**

| Author | Distribution | N | Mean and/or median (days) with 95%CI | Plausible range (days) with 95%CI |
|---|---|---|---|---|
| Our study | Weibull | 19 | Mean 6.4 (4.9–8.2) | 1.4–13.0 (2.5th to 97.5th percentile) |
| | | | Median 6.1 (4.4–8.0) | |
| Backer et al. [8] | Weibull | 88 | Mean 6.4 (5.6–7.7) | 2.1–11.1 days (2.5 to 97.5 percentile) |
| | | | Median 6.4 (5.5–7.5) | |
| Lauer et al. [7] | Lognormal | 181 | Median 5.2 (4.4–6.0) | 2.2–11.5 (2.5th to 97.5th percentile) |
| Linton et al. [12] | Lognormal | 58 (excluding Wuhan residents) | *Non-truncated* | 95th percentile for non- truncated data: 10.6 (8.5–14.1) |
| | | | Mean 5.0 (4.2–6.0), | |
| | | | Median 4.3 (3.5–5.1) | |
| | | | *Right-truncated* | 95th percentile for right-truncated: 12.3 (9.1–19.8) |
| | | | Mean 5.6 (4.4–7.4) | |
| | | | Median 4.6 (3.7–5.7) | |
| | Lognormal | 152 (including Wuhan residents) | Nontruncated: | 95th percentile for non-truncated data 10.8 (95% CI: 9.3–12.9) |
| | | | Mean 5.6 (5.0–6.3), Median 5.0 (95%: 5.0–6.3) | |
| Li et al. [11] | Lognormal | 10 | Mean 5.2 (4.1–7.0), | 95th percentile"12.5 (9.2–18) |
| Leung [13] | Weibull | 175 (Travelers to Hubei) | Mean 1.8 (1.0–2.7) | 95th percentile 3.2 (1.0–3.8) |
| | Weibull | 175 (Non-travelers) | Mean 7.2 (7.1–8.4) | 95th percentile 14.6 (12.1–17.1) |
| Jiang et al. [9] | | 50 | Mean 4.9 (4.4–5.5) | |
| Lee et al. [10] | Lognormal | 47 | Median 3 (0.6–8.2) | |
| Kong [11] | | 136 | Mean 8.5 (7.8–9.2) | |
| | | | Median 8.3 (7.6–9.0) | |
| Tindale et al. [12] | | | Mean 5.99 (95%CI 4.97, 7.14) | |
| | | | Median 5.32 | |
| | | | Tianjin | |
| | | | Mean 8.68 (7.72, 9.7) | |
| | | | Median 8.06; | |
| Qin et al. [13] | | 1211 | Mean 8·62 (95% CI: 8·02–9·28), | |
| | | | median 8·13 days (7·37–8·91), | |

The estimated mean of incubation period for 19 Vietnamese confirmed COVID-19 cases using Weibull distribution model is higher than that of SARS in Hong Kong and Beijing [21] and in MERS [15, 16, 22, 23]. Hence, mathematical models of COVID-19 should not be fitted with incubation period of MERS or SARS.

The variation of incubation period of SARS-CoV-2 could be explained by the differences on the strains of the virus, the biological variations of population and the control measures of certain country. It is suggested that any interpretation of results on incubation time of SARS-CoV-2 is strongly dependent on the selected data sources [9]. In Vietnam, the key COVID-19 control strategy has focused on active case finding, early testing, treating and strictly isolating cases, which has been considered as effective in finding people with SARS-CoV-2 positive and providing treatment prior to clinical symptoms presentation [16]. It is also a possible circumstance that people with suggestive symptoms of respiratory diseases may take drugs before seeking diagnosis and treatment.

In this study, the plausible range of incubation period estimated using the Weibull distribution supports the importance to isolate confirmed and suspected cases, and close contacts for 14 days after exposure. However, we recommend the policy makers to consider the upper bound of this range (97.5th percentile 13.0, CI 10.6–20.7). Prior studies also suggested that the

incubation period in some cases can be up to 24 days [24]. As such, the quarantine period should be extended up to three weeks.

Our study has several strengths, including being the first effort to estimate SAR-CoV-2 incubation period using data from Vietnamese locally transmitted cases. We also used 3 different models for estimation and identified the model that best fitted with our data. However, there are potential limitations in our study. First, not all confirmed COVID-19 cases have completed epidemiological data on exposure interval and date of symptom onset available on public government reports. Second, the number of patients included in the analysis is small, possibly explaining the wider range of estimated incubation period. In addition, the study could not analyze the differences in incubation period between age groups or gender due to small sample size. The large variation in incubation periods in our study can also be attributed by variations in exposure period and uncertainties in the date of infection. These highlight the need for data standardization for further studies on COVID-19 and the importance of publishing data for the scientific and public health community [9].

This study may contribute to the effort of COVID-19 control effort in Vietnam and elsewhere by providing an informed estimate of the incubation period. This is a key variable needed for modelling the spread of SARS-CoV-2, for informed decision-making throughout the pandemic. Similar methodology could be used to estimate the incubation period for other countries and diseases.

## Author Contributions

**Conceptualization:** Long V. Bui, Ha T. Nguyen, Toan T. T. Do, Ngoc T. Pham, My Hanh Bui.

**Data curation:** Thu-Anh Nguyen, Thuy P. Nguyen, Truong T. Nguyen, Toan T. T. Do, My Hanh Bui.

**Formal analysis:** Long V. Bui, Ha T. Nguyen, Hagai Levine, Ha N. Nguyen, Thu-Anh Nguyen, Thuy P. Nguyen, Truong T. Nguyen, Toan T. T. Do, Ngoc T. Pham, My Hanh Bui.

**Investigation:** Long V. Bui, Ha T. Nguyen, Ha N. Nguyen, Thu-Anh Nguyen, Thuy P. Nguyen, Toan T. T. Do, Ngoc T. Pham, My Hanh Bui.

**Methodology:** Long V. Bui, Ha T. Nguyen, Hagai Levine, Ha N. Nguyen, Thu-Anh Nguyen, Thuy P. Nguyen, Truong T. Nguyen, Toan T. T. Do, Ngoc T. Pham, My Hanh Bui.

**Project administration:** Long V. Bui, Ha T. Nguyen, My Hanh Bui.

**Resources:** Long V. Bui, Ha T. Nguyen, Ha N. Nguyen, Thu-Anh Nguyen, Truong T. Nguyen, Ngoc T. Pham, My Hanh Bui.

**Software:** Long V. Bui, Hagai Levine, Ha N. Nguyen, Thu-Anh Nguyen, Truong T. Nguyen, Ngoc T. Pham, My Hanh Bui.

**Supervision:** Long V. Bui, Ha T. Nguyen, My Hanh Bui.

**Validation:** Long V. Bui, Ha T. Nguyen, Thu-Anh Nguyen, My Hanh Bui.

**Visualization:** Hagai Levine, Ha N. Nguyen, Thu-Anh Nguyen, Thuy P. Nguyen, Truong T. Nguyen, Toan T. T. Do.

**Writing – original draft:** Long V. Bui, Hagai Levine, Ha N. Nguyen, Thuy P. Nguyen, Truong T. Nguyen, Toan T. T. Do, Ngoc T. Pham, My Hanh Bui.

**Writing – review & editing:** Long V. Bui, Ha T. Nguyen, My Hanh Bui.

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
