## [Decision Letter · Decision Letter 0]

25 Aug 2020

PONE-D-20-14137

Estimation of the incubation period of SARS-CoV-2 in Vietnam

PLOS ONE

Dear Dr. Bui,

Thank you for submitting your manuscript to PLOS ONE. After careful consideration, we feel that it has merit but does not fully meet PLOS ONE’s publication criteria as it currently stands. Therefore, we invite you to submit a revised version of the manuscript that addresses the points raised during the review process.

We look forward to receiving your revised manuscript.

Kind regards,

Ka Chun Chong

Academic Editor

PLOS ONE

Journal Requirements:

Additional Editor Comments :

As there are a number of studies related with an estimation of the incubation period, a review of the estimates in the literature is required.

Reviewers' comments:

Reviewer's Responses to Questions

**Comments to the Author**

1. Is the manuscript technically sound, and do the data support the conclusions?

Reviewer #1: Yes

Reviewer #2: Partly

2. Has the statistical analysis been performed appropriately and rigorously? 

Reviewer #1: Yes

Reviewer #2: I Don't Know

3. Have the authors made all data underlying the findings in their manuscript fully available?

Reviewer #1: Yes

Reviewer #2: Yes

4. Is the manuscript presented in an intelligible fashion and written in standard English?

Reviewer #1: Yes

Reviewer #2: Yes

5. Review Comments to the Author

Reviewer #1: The manuscript is well written and has a practical implications for addressing the COVID-19 response in Vietnam. The authors use small number of confirmed cases to estimate the incubation period and I have some comments for this manuscript.

Minor comments

1. The analyses were highly dependent on the epidemiological investigations for each case. Please describe the process of the data generation such as epidemiological investigations briefly to show the validity of the data such as the date of exposure and illness onset.

2. The author compared the incubation period in other studies using the data from China. Can you please add the other studies using the data from out of china? For example, Korea (Lee H et al. Journal of Infection and Chemotherapy 2020), France (Slje H, hal-pasteur.achieve 2020), etc).

3. Please provide more detail of the limitation of this study, particularly on the small sample size to analyse.

Reviewer #2: The authors estimated the incubation period of Vietnamese confirmed COVID-19 cases from a subsample of 19 of the 102 locally infected cases for the period 23 Jan to 13 Apr 2020. My review comments are:

1. How representative is the subsample? Data on age range and sex of the subsample was mentioned in the article, but not for the parent sample of 102 patients. Were their age/sex data comparable?

2. The authors attributed the large variation in incubation periods in their study to small sample size, but can these be contributed also by variations in exposure period and uncertainties in the date of infection (Figure 2)?

3. The authors commented that “Only one study by Leung [13] estimated longer incubation periods than our estimation”. But there are three other studies that gave estimates of longer mean incubation periods of COVID-19 in the order of 8 days:

Kong T-K. Longer incubation period of coronavirus disease 2019 (COVID-19) in older adults. Aging Medicine 2020;3:102-109.

Qin J, You C, Lin Q, Hu T, Yu S, Zhou X-H. Estimation of incubation period distribution of COVID-19 using disease onset forward time: a novel cross-sectional and forward follow-up study. Sci Adv. 2020. https://advances.sciencemag.org/content/early/2020/08/07/sciadv.abc1202

Tindale LC, Stockdale JE, Coombe M, et al. Evidence for transmission of COVID-19 prior to symptom onset. eLife 2020;9:e57149. https://doi.org/10.7554/eLife.57149

4. The authors recommended the quarantine period should be extended up to three weeks based on their study finding of 97.5th percentile 13.0, CI 10.6–20.7. But would they also comment on the wide CI of 10.6-20.7 and the reason for the uncertainty?

6. PLOS authors have the option to publish the peer review history of their article (what does this mean?). If published, this will include your full peer review and any attached files.

Reviewer #1: No

Reviewer #2: No

---

## [Author Response · Author response to Decision Letter 0]

20 Oct 2020

Dear Reviewer, 

Thank you for your helpful comments. We have revised our paper accordingly and feel that your comments helped clarify and improve our paper. Please find our response (in blue) to reviewer’s specific comments (in black) below.

Reviewer #1:

1. The analyses were highly dependent on the epidemiological investigations for each case. Please describe the process of the data generation such as epidemiological investigations briefly to show the validity of the data such as the date of exposure and illness onset.

Response: Thank you for your comment. I have added details on the data generation: According to the report flow of the Ministry of Health and National Steering Committee for COVID-19 response, information of all laboratory-confirmed COVID-19 cases in Vietnam, which include patient number, travelling history, contact tracing and clusters, date of laboratory-confirmed, date of assertion, date and status of discharge was collected by the local centers for disease control and hospitals where patients were being admitted. The information then was reported daily to the Ministry of Health and officially made publicly available on the website httpp://ncov.moh.gov.vn. 

This method was also used by several articles on COVID-19 in Vietnam, which were cited in our manuscript.

2. The author compared the incubation period in other studies using the data from China. Can you please add the other studies using the data from out of china? For example, Korea (Lee H et al. Journal of Infection and Chemotherapy 2020), France (Slje H, hal-pasteur.achieve 2020), etc).

Reponse: Thank you very much. We have added relevant citation as you suggested. We did not cite Slje H, hal-pasteur.achieve 2020 because the article did not estimate the incubation period of COVID-19 in France, instead, the author used the incubation period of 5 days in a deterministic compartmental model stratified by age to describe the transmission of SARSCoV-2 in the French population.

3. Please provide more detail of the limitation of this study, particularly on the small sample size to analyse.

Response: Thank you very much. We have added “In addition, the study could not analyze the differences in incubation period between age groups due to small sample size”.

Reviewer #2:

1. How representative is the subsample? Data on age range and sex of the subsample was mentioned in the article, but not for the parent sample of 102 patients. Were their age/sex data comparable?

Response: Thank you very much for your comment. We have added some lines that compare the age/sex of subgroup to the parent sample. “There was no statistical difference in mean age and sex distribution between sub-group for analysis and the entire group (p<0.05).”

2. The authors attributed the large variation in incubation periods in their study to small sample size, but can these be contributed also by variations in exposure period and uncertainties in the date of infection (Figure 2)?

Response: Thank you very much. We’ve added this issue in the limitations: “The large variation in incubation periods in our study can also be attributed by variations in exposure period and uncertainties in the date of infection”.

3. The authors commented that “Only one study by Leung [13] estimated longer incubation periods than our estimation”. But there are three other studies that gave estimates of longer mean incubation periods of COVID-19 in the order of 8 days: Kong T-K. Longer incubation period of coronavirus disease 2019 (COVID-19) in older adults. Aging Medicine 2020;3:102-109.

Qin J, You C, Lin Q, Hu T, Yu S, Zhou X-H. Estimation of incubation period distribution of COVID-19 using disease onset forward time: a novel cross-sectional and forward follow-up study. Sci Adv. 2020. https://advances.sciencemag.org/content/early/2020/08/07/sciadv.abc1202

Response: Thank you very much. We have added relevant citation as you suggested. We did not cite (Slje H, hal-pasteur.achieve 2020) because the article did not estimate the incubation period of COVID-19 in France, instead, the author used the incubation period of 5 days in a deterministic compartmental model stratified by age to describe the transmission of SARSCoV-2 in the French population.

4. The authors recommended the quarantine period should be extended up to three weeks based on their study finding of 97.5th percentile 13.0, CI 10.6–20.7. But would they also comment on the wide CI of 10.6-20.7 and the reason for the uncertainty?

Response: Thank you very much. We’ve added one reference, suggesting that the incubation period of CVOID-19 may be up to 24 days “Prior studies also suggested that the incubation period in some cases can be up to 24 days [24]”.

Thank you again for your constructive manuscript, that absolutely improve our manuscript.

We hope reviewers take our revised version into your consideration.

We look forward to hearing positive feedback from you.

Sincerely Yours

Bui My Hanh

---

## [Decision Letter · Decision Letter 1]

12 Nov 2020

PONE-D-20-14137R1

Estimation of the incubation period of COVID-19 in Vietnam

PLOS ONE

Dear Dr. Bui,

Thank you for submitting your manuscript to PLOS ONE. After careful consideration, we feel that it has merit but does not fully meet PLOS ONE’s publication criteria as it currently stands. Therefore, we invite you to submit a revised version of the manuscript that addresses the points raised during the review process.

ACADEMIC EDITOR: Please address the remaining comments.

We look forward to receiving your revised manuscript.

Kind regards,

Ka Chun Chong

Academic Editor

PLOS ONE

Reviewers' comments:

Reviewer's Responses to Questions

**Comments to the Author**

1. If the authors have adequately addressed your comments raised in a previous round of review and you feel that this manuscript is now acceptable for publication, you may indicate that here to bypass the “Comments to the Author” section, enter your conflict of interest statement in the “Confidential to Editor” section, and submit your "Accept" recommendation.

Reviewer #1: All comments have been addressed

Reviewer #2: All comments have been addressed

2. Is the manuscript technically sound, and do the data support the conclusions?

Reviewer #1: Yes

Reviewer #2: Yes

3. Has the statistical analysis been performed appropriately and rigorously? 

Reviewer #1: Yes

Reviewer #2: Yes

4. Have the authors made all data underlying the findings in their manuscript fully available?

Reviewer #1: Yes

Reviewer #2: Yes

5. Is the manuscript presented in an intelligible fashion and written in standard English?

Reviewer #1: Yes

Reviewer #2: Yes

6. Review Comments to the Author

Reviewer #1: The authors reflect all of the comments in the revised manuscript; The readibility is improved and looks better.

Reviewer #2: My review comments have been addressed and the manuscript is acceptable for publication after correction of one error and a typo:

Line 157 error: Lee et al [10] should not be quoted as a study showing longer incubation period than the authors' study finding, because the median incubation period of Lee et al [10] is shorter at 3 days. It can nevertheless be quoted under Line 155 together with other studies showing shorter incubation period compared with the authors' finding.

Line 112 typo: "of those" is duplicated.

7. PLOS authors have the option to publish the peer review history of their article (what does this mean?). If published, this will include your full peer review and any attached files.

Reviewer #1: No

Reviewer #2: No

---

## [Author Response · Author response to Decision Letter 1]

17 Nov 2020

Dear Reviewers, 

Thank you for your helpful comments. We have revised our paper accordingly and feel that your comments helped clarify and improve our paper. Please find our response to reviewer’s specific comments below.

Reviewer #1:

The authors reflect all of the comments in the revised manuscript; The readibility is improved and looks better.

Response: Thank you for your comment.

Reviewer #2: 

My review comments have been addressed and the manuscript is acceptable for publication after correction of one error and a typo:

Line 157 error: Lee et al [10] should not be quoted as a study showing longer incubation period than the authors' study finding, because the median incubation period of Lee et al [10] is shorter at 3 days. It can nevertheless be quoted under Line 155 together with other studies showing shorter incubation period compared with the authors' finding.

Line 112 typo: "of those" is duplicated.

Response: Thank you for your comment. We have deleted the duplicated “of those”. We have moved the reference of Lee et al. to studies estimating shorter incubation time than ours and revised the range of shorter estimation from 3 – 5.6 days.

We have made overall editing to the manuscript to make sure all typos and wordings are correct. 

Thank you again for your constructive manuscript, that absolutely improve our manuscript.

We hope reviewers take our revised version into your consideration.

We look forward to hearing positive feedback from you.

Sincerely Yours

Bui My Hanh

---

## [Editor Report · Decision Letter 2]

1 Dec 2020

Estimation of the incubation period of COVID-19 in Vietnam

PONE-D-20-14137R2

Dear Dr. Bui,

We’re pleased to inform you that your manuscript has been judged scientifically suitable for publication and will be formally accepted for publication once it meets all outstanding technical requirements.

Kind regards,

Ka Chun Chong

Academic Editor

PLOS ONE
---

## [Editor Report · Acceptance letter]

10 Dec 2020

PONE-D-20-14137R2 

Estimation of the incubation period of COVID-19 in Vietnam 

Dear Dr. Bui:

I'm pleased to inform you that your manuscript has been deemed suitable for publication in PLOS ONE. Congratulations! Your manuscript is now with our production department. 

Kind regards, 

on behalf of

Dr. Ka Chun Chong 

Academic Editor

PLOS ONE